# Myrcene Salvages Rotenone-Induced Loss of Dopaminergic Neurons by Inhibiting Oxidative Stress, Inflammation, Apoptosis, and Autophagy

**DOI:** 10.3390/molecules28020685

**Published:** 2023-01-10

**Authors:** Sheikh Azimullah, Richard L. Jayaraj, Mohamed Fizur. Nagoor Meeran, Fakhreya Y. Jalal, Abdu Adem, Shreesh Ojha, Rami Beiram

**Affiliations:** 1Department of Pharmacology and Therapeutics, College of Medicine and Health Sciences, United Arab Emirates University, Al Ain P.O. Box 15551, United Arab Emirates; 2Department of Pharmacology, College of Medicine and Health Sciences, Khalifa University, Abu Dhabi P.O. Box 127788, United Arab Emirates

**Keywords:** Parkinson’s disease, rotenone, myrcene, inflammation, oxidative stress, autophagic flux

## Abstract

Parkinson’s disease (PD) is characterized by the loss of dopaminergic neurons in the substantia nigra pars compacta, resulting in motor deficits. The exact etiology of PD is currently unknown; however, the pathological hallmarks of PD include excessive production of reactive oxygen species, enhanced neuroinflammation, and overproduction of α-synuclein. Under normal physiological conditions, aggregated α-synuclein is degraded via the autophagy lysosomal pathway. However, impairment of the autophagy lysosomal pathway results in α-synuclein accumulation, thereby facilitating the pathogenesis of PD. Current medications only manage the symptoms, but are unable to delay, prevent, or cure the disease. Collectively, oxidative stress, inflammation, apoptosis, and autophagy play crucial roles in PD; therefore, there is an enormous interest in exploring novel bioactive agents of natural origin for their protective roles in PD. The present study evaluated the role of myrcene, a monoterpene, in preventing the loss of dopaminergic neurons in a rotenone (ROT)-induced rodent model of PD, and elucidated the underlying mechanisms. Myrcene was administered at a dose of 50 mg/kg, 30 min prior to the intraperitoneal injections of ROT (2.5 mg/kg). Administration of ROT caused a considerable loss of dopaminergic neurons, subsequent to a significant reduction in the antioxidant defense systems, increased lipid peroxidation, and activation of microglia and astrocytes, along with the production of pro-inflammatory cytokines (IL-6, TNF-α, IL-1β) and matrix metalloproteinase-9. Rotenone also resulted in impairment of the autophagy lysosomal pathway, as evidenced by increased expression of LC3, p62, and beclin-1 with decreased expression in the phosphorylation of mTOR protein. Collectively, these factors result in the loss of dopaminergic neurons. However, myrcene treatment has been observed to restore antioxidant defenses and attenuate the increase in concentrations of lipid peroxidation products, pro-inflammatory cytokines, diminished microglia, and astrocyte activation. Myrcene treatment also enhanced the phosphorylation of mTOR, reinstated neuronal homeostasis, restored autophagy-lysosomal degradation, and prevented the increased expression of α-synuclein following the rescue of dopaminergic neurons. Taken together, our study clearly revealed the mitigating effect of myrcene on dopaminergic neuronal loss, attributed to its potent antioxidant, anti-inflammatory, and anti-apoptotic properties, and favorable modulation of autophagic flux. This study suggests that myrcene may be a potential candidate for therapeutic benefits in PD.

## 1. Introduction

Parkinson’s disease (PD) is a common aging-related neurodegenerative disease is clinically characterized by impaired motor function along with symptoms including bradykinesia, rigidity, tremor, postural changes, cognitive abnormalities, emotional, and olfactory alterations [1,2]. Intracytoplasmic accumulation of α-synuclein protein, known as Lewy bodies, and dopaminergic neuronal loss in the substantia nigra pars compacta (SN) area are prominent clinical signs of PD [3,4]. The exact mechanism of PD is still unknown; however, it is characterized by degeneration of dopaminergic neurons in the SN. Many factors, including oxidative stress, inflammation, mitochondrial dysfunction, apoptosis, and autophagy, play an important pathogenic role in the causation and progression of PD [5,6].

Oxidative stress involves the overgeneration of reactive oxygen species (ROS), which cause excessive lipid and protein oxidation, mitochondrial dysfunction, reduced kinase activity, and protein misfolding, leading to the loss of dopaminergic neurons [7,8,9,10]. Oxidative stress, in turn, activates glial cells, resulting in the induction of pro-inflammatory mediators that cause neuroinflammation [11,12]. Low-grade chronic neuroinflammation plays an important pathological role in dopaminergic neurodegeneration and leads to PD [13,14,15,16,17]. Additionally, tissue inhibitor metalloproteinases (TIMPs) of matrix metalloproteinases (MMPs), including MMP-9, were found to be elevated in the cerebrospinal fluid of individuals with PD [5]. Autophagy is a catabolic mechanism that removes unnecessary or dysfunctional proteins and damaged organelles via lysosomal machinery [18,19,20]. Impaired autophagy leads to the accumulation of α-synuclein, and the resultant proteotoxicity plays a critical role in PD pathogenesis. Under normal conditions, growth, survival, proliferation, and cellular homeostasis are maintained by the mTOR signaling pathway. Decreased mTOR, following phosphorylation of the p70S6 kinase pathway, leads to neuronal loss [21].

Considering the role of oxidative and nitrative stress, inflammation, apoptosis, and autophagy in triggering dopaminergic neurodegeneration, approaches to tackle these events using bioactive agents of natural origin are gaining attention [22,23,24,25,26,27]. Agents that can mitigate oxidative stress, inflammation, autophagy, and rescue neurons could be vital in delaying, preventing, or treating PD. Among the treasures of bioactive agents from nature, terpenes and terpenoids, specifically present in plants providing aroma due to essential oils, have garnered attention for their neuroprotective properties and dietary availability. Myrcene, a dietary monoterpene abundantly present in many edible plants, including lemongrass, mangoes, cannabis, verbena, bay, and citrus fruits, is popularly used in food, cosmetics, and beverages as a food additive, and is generally considered safe. It has garnered significant attention for its potential health benefits and pharmacological properties, including analgesic [28], antihyperglycemic [29], antioxidant, anti-inflammatory [30], and antibacterial [31]. The neuroprotective effects of myrcene are attributed to its powerful antioxidant, anti-inflammatory, and antiapoptotic properties [32]. Existing reports related to its neuroprotective effects against Alzheimer’s disease, cerebral ischemia-reperfusion injury, and neuropathy, along with its dietary availability and safety, have encouraged evaluation of its role in dopaminergic neurodegeneration.

To induce dopaminergic neurodegeneration, we employed a rodent model of PD induced by rotenone (ROT), a plant-extracted chemical pesticide/insecticide. It is an environmental toxin and is widely known to induce biochemical, molecular, and pathological changes, as seen in PD, which is ascribed to its mitochondrial complex-1 inhibitor properties [33]. ROT is a highly lipophilic molecule that crosses the blood–brain barrier, resulting in ATP loss, increased oxidative stress, and activation of microglia and astrocytes, resulting in the induction of pro-inflammatory cytokines and MMPs, and eventually dopaminergic neurodegeneration [34,35]. Therefore, in the present study, we evaluated the effect of myrcene on ROT-induced dopaminergic neurodegeneration, mimicking the features of PD. The underlying mechanisms were investigated to determine the parameters of oxidative stress, inflammation, apoptosis, and autophagy.

## 2. Results

### 2.1. Myrcene Diminished ROT-Induced Oxidative Stress in Rats

The effects of myrcene on oxidative stress markers are shown in Figure 1. ROT-injected rats showed a significant increase in MDA (F 3,19 = 11.223 *p* < 0.001) and decrease in GSH (F 3,19 = 11.020 *p* < 0.001) levels in the mid brain compared to the control group. However, myrcene treatment significantly reduced MDA levels and significantly (*p* < 0.05) increased GSH levels in ROT-injected rats. ROT injections also induced a significant reduction in SOD (F 3,22 = 13.578 *p* < 0.001) and catalase (F 3,20 = 6.935 *p* < 0.001) activities, in comparison with the control group. Similar to the improvement in GSH levels, myrcene treatment significantly (*p* < 0.05) improved SOD and catalase activities in ROT-injected rats.

### 2.2. Myrcene Downregulated ROT-Induced Pro-Inflammatory Cytokines, MMP-9, Inflammatory Mediators, and Apoptosis

The effects of myrcene on pro-inflammatory cytokines (IL-6, IL-1β, TNF-α) and MMP-9 are shown in Figure 2. ROT injections caused a significant increase in the pro-inflammatory cytokines IL6 (F 3,17 = 32.114 *p* < 0.001), IL-1β (F 3,18 = 22.025 *p* < 0.001), TNF-α (F 3,18 = 27.025 *p* < 0.001), and MMP-9 (F 3,18 = 25.033 *p* < 0.001) levels compared to the control group. However, myrcene treatment significantly reduced the pro-inflammatory cytokines and MMP-9 levels compared to those in rats injected with ROT. In addition to pro-inflammatory cytokines, iNOS and COX-2 were also detected in midbrain samples (Figure 2A). ROT injections induced a significant increase in iNOS and COX-2 expression compared to the control group. However, myrcene treatment caused a significant decrease in iNOS (F 3,24 = 445.564 *p* < 0.001) and COX-2 (F 3,24 = 38.779 *p* < 0.001) expression (Figure 2. Additionally, the significant effect of myrcene on apoptotic cell death (markers of apoptosis, Bax (F 3,24 = 18.310 *p* < 0.001), Bcl-2 (F 3,24 = 5.6 *p* < 0.001) and BAX/Bcl-2 ratio) (F 3,24 = 15.230 *p* < 0.001) in dopaminergic neurons was determined (Figure 2). ROT injections increased the expression of pro-apoptotic protein Bax and decreased the expression of anti-apoptotic protein Bcl-2. However, treatment with myrcene inhibited apoptotic cell death, as indicated by the reduced expression of Bax, BAX/Bcl-2 ratio, and increased expression of Bcl-2 in the striatum of the ROT-injected rats (Figure 2B)

### 2.3. Myrcene Prevented ROT-Induced Loss of Dopaminergic Neurons

The loss of TH-positive neurons in the SN and the subsequent reduction in striatal TH expression are characteristic pathogenic processes in PD. Therefore, we assessed the number of TH +ve dopaminergic neurons in the SN, as well as their expression in the striatum. ROT injections into the rats resulted in the loss of TH-positive neurons (F 3, 20 = 35.100 *p* < 0.000), represented by a 50% decrease in the intensity of TH +ve striatal fibers (Figure 3). Interestingly, myrcene treatment in rats injected with ROT prevented damage to dopaminergic neurons and increased TH (F 3, 20 = 11.223 *p* < 0.000) expression in striatal fibers, as demonstrated by immunohistochemical analysis.

### 2.4. Myrcene Reduced the ROT-Induced Activation of Microglia and Astroglia

Figure 4 reveals Iba-1 and GFAP, a marker of reactive microglia and astrocytes, respectively, using immunofluorescent labeling in the striatum. The quantitative analysis of activated microglia and astrocytes is shown as a percentage of the control (Figure 4). ROT injections trigger microglial activation (F 3,24 = 7.422 *p* < 0.000) and appear with large cell bodies and lesser processes along with activated astrocytes, as demonstrated by the increased expression (F 3,19 = 9.11 *p* < 0.000) of GFAP-positive cells. Interestingly, myrcene treatment substantially reduced Iba-1 and GFAP expression in the striatum, compared with that in ROT-injected rats. 

### 2.5. Myrcene Mitigated ROT-Induced Increased Expression of α-Synuclein and Autophagy

Accumulation of α-synuclein protein in Lewy bodies is a major pathogenic occurrence in PD, and increased α-synuclein expression causes neuronal death via necrosis or apoptosis. Furthermore, autophagy dysfunction leads to the formation of aggregated α-synuclein proteins, which promotes neurodegeneration. Western blot analysis revealed that ROT injections resulted in a substantial increase (three-fold) in α-synuclein expression in the striatum (Figure 5). However, myrcene treatment considerably reduced the expression of α-synuclein. These findings suggest that myrcene lowers α-synuclein expression (F 3,24 = 5.6 *p* < 0.000), which is attributed to the preservation of dopaminergic neurons. Furthermore, ROT injections induced a substantial increase in autophagosome accumulation, as depicted by the increase in beclin-1 (F 3,24 = 5.4 *p* < 0.000) and LC3II/LC3I ratio (F 3,24 = 7.8 *p* < 0.001). Myrcene treatment in ROT-injected rats decreased the expressions of beclin-1 and LC3II/LC3I ratio, indicating a reduction in autophagosome accumulation (Figure 6). To investigate the role of myrcene in the regulation of autophagy, the expression of p62 (F 3,24 = 9.8 *p* < 0.001), which connects ubiquitinated proteins with the autophagy process via beclin-1 and LC3, was examined. ROT injections resulted in a substantial increase in p62 expression, indicating inhibition of autophagic clearance by ROT. Myrcene treatment resulted in a substantial decrease in p62 expression, indicating autophagic clearance of misfolded proteins by myrcene (Figure 6). Furthermore, neither the ratio of LC3II/LC3I nor the level of p62 were altered substantially in the control or myrcene alone-treated groups.

### 2.6. Myrcene Modulated ROT-Induced mTOR Signaling Pathway

Several studies have found that mTOR inhibition causes neuronal dysfunction and negatively impacts neuronal regeneration mechanisms. In the central nervous system, aggregated protein buildup boosts the activity of mTOR (F 3,24 = 2.5 *p* < 0.001) at first, but gradually decreases thereafter, leading to impairment in neuronal homeostasis. ROT injections resulted in a substantial decrease in mTOR phosphorylation (F3,19 = 16.35 *p* < 0.001), according to immunoblotting analyses in the striatum (Figure 6. However, myrcene treatment restored mTOR signaling by increasing mTOR phosphorylation. Our findings imply that myrcene inhibits ROT-mediated neuronal impairment by reinstating the mTOR signaling pathway.

## 3. Discussion 

These findings demonstrate that myrcene mitigates dopaminergic neuronal loss due to its antioxidant, anti-inflammatory, and antiapoptotic properties, along with favorable modulation of autophagic flux. Additionally, the dose of myrcene used in this study was not associated with neuronal toxicity.

ROT-induced rodent models of PD are widely used to understand the pathogenesis of neurodegeneration, and evaluate the therapeutic potential of natural compounds [26,27]. PD is characterized by increased ROS, upregulated pro-inflammatory cytokines, and MMPs, resulting in dopaminergic neuronal loss by apoptosis or excessive autophagy [25,36,37]. ROT is a lipophilic insecticide that crosses the blood–brain barrier [38]. ROT accumulates in the mitochondria and mediates the inhibition of mitochondrial complex 1, thereby increasing ROS and pro-inflammatory factors [27,39,40]. Drugs that mitigate the burden of oxidative stress and inflammation may be potential therapeutic candidates for PD. 

In recent years, numerous phytochemicals have been evaluated in PD models, and myrcene, a monoterpene, has attracted considerable attention because of its multiple pharmacological properties [29,31,41]. ROT, a mitochondrial complex-1 inhibitor, suppresses mitochondrial function associated with ATP loss and increases ROS production [5,42]. ROS intensification causes lipid peroxidation which triggers MDA formation, a key factor in PD pathogenesis [5,11,37]. Brain tissues are extremely vulnerable to oxidative injuries because of the high amount of fatty acids, resulting in the subsequent decrease in enzymatic (SOD, catalase) and non-enzymatic (GSH) antioxidant defense systems [43]. The disparity between the endogenous antioxidant system and free radical-induced oxidative damage is associated with increased MDA and decreased SOD, catalase, and GSH levels at the cellular level. The protective effect of myrcene against ROT-induced pathological changes observed in our study clearly demonstrated its capacity as a powerful antioxidant against ROT-induced neuronal damage. 

It is well established that oxidative stress and neuroinflammation accompany each other during the progression of PD. Oxidative stress acts as an activator of many pathological mechanisms, such as neuroinflammation, autophagy, apoptosis, and impairment of cellular homeostasis [25,44,45,46,47]. Experimental and clinical studies have demonstrated that inflammation aggravates neurodegeneration in PD. Sustained chronic inflammation is a key pathological event in many neurodegenerative diseases including PD [48]. In PD, microglia and astrocytes are activated and secrete pro-inflammatory cytokines (TNF-α, IL-1β, and IL6). Myrcene treatment successfully counters ROT-induced changes in the activities of pro-inflammatory cytokines, clearly demonstrating its potent anti-inflammatory effect. 

The activation of microglia and astrocytes leads to a cycle of inflammatory events that fuel the progressive loss of dopaminergic neurons in PD brains. Furthermore, in the striatum, there is a considerable increase in the expression of inflammatory mediators (iNOS and COX-2) [17]. The induction of pro-inflammatory cytokines enhances iNOS production and exacerbates nitric oxide-induced dopaminergic neuronal damage. Subsequently, pro-inflammatory cytokines accentuate activation of microglia and subsequent infiltration across the blood–brain barrier, which further leads to chronic activation of resident microglia and the resultant chronic inflammatory response. These persistent inflammatory responses trigger PD progression [4]. ROT challenge results in increased iNOS and COX-2 expression in the brain. Higher levels of tissue inhibitors of MMPs, including MMP-9, have been reported in the cerebrospinal fluid of patients with PD [5]. Activated microglia and astrocytes are the main inducers of MMPs, including MMP-9 a chief element of the basement membrane that contributes to the onset and progression of neurodegenerative diseases, including PD. Interestingly, myrcene treatment lowered expressions/activities of iNOS, COX-2, and MMP-9, again revealing its potent anti-inflammatory properties. 

TH is a rate-limiting enzyme that catalyzes the formation of L-DOPA during dopamine (DA) biosynthesis. TH deficiency in the striatum is an important characteristic of PD. DA, a neurotransmitter, is synthesized in dopaminergic neurons in the SN [1]. Enzyme activity generates ROS in vitro, and is a target for radical-mediated oxidative damage in TH pathogenesis. In addition, dopaminergic neurons are lost, followed by withdrawal of dopaminergic nerve terminals in the striatum owing to DA deficiency [6]. Consistent with these reports, our results showed that ROT injections caused the death of dopaminergic neurons, with reduced TH-ir fibers in the striatum. However, myrcene treatment reinstated dopaminergic neuronal death after ROT insult, and clearly showed an anti-neuroprotective effect (Figure 3).

The pathological alteration of α-synuclein in ROT-induced rats is an important hallmark of PD. The exact mechanism of α-synuclein in PD is not known; however, dopaminergic neuronal loss in SN is associated with aberrant changes in α-synuclein. Various reports have shown that α-synuclein increased expression activates microglia, thus initiating the pathogenesis of PD through the production of ROS by triggering apoptosis [2,36,49]. Collectively, the present study is in line with previous findings that an increase in α-synuclein expression occurs after ROT-insult. However, a reduction in the expression of α-synuclein clearly indicates myrcene protective ability in reducing dopaminergic neurodegeneration by reducing α-synuclein expression. 

Apoptosis induced by various factors, such as ROS and aberrant α-synuclein aggregation, is a key process involved in PD. Apoptosis is an evolutionary process that is detrimental to dopaminergic neurons in the SN and striatum in PD [50,51,52]. Dispasquale (1991), for the first time, reported both mitochondrial complex-1 inhibition and α-synuclein-mediated apoptosis and associated neuronal death. α-Synuclein co-exists with the anti-apoptotic neuronal factor Bcl-2 and inhibits its expression [52,53,54]. The balance between bcl2 and Bax proteins is related to cell viability. Bax-induced alterations in mitochondrial membrane potential trigger the release of cytochrome C into the cytosol, which then activates caspases, resulting in cell death [2]. Myrcene favorably regulates the Bax/Bcl2 balance and secures dopaminergic neurons by preventing apoptosis, clearly revealing its potent anti-apoptotic effects. 

As previously stated, the build-up of α-synuclein, due to impaired autophagic processes, plays a critical role in the development of Lewy bodies and subsequent neurodegeneration in PD. Additionally, ROS are key factors in mediating neuronal metabolism and autophagy [27]. Autophagy is an evolutionarily conserved catabolic process that facilitates the degradation of long-lasting proteins and impaired or unessential organelles back into the cytoplasm for cell homeostasis [55]. In PD, autophagy is influenced by various factors, such as increased expression of α-synuclein, ROS overproduction, release of pro-inflammatory cytokines, and growth nutrient impairment. The autophagy substrate molecule p62 links ubiquitinated proteins with LC3 for autophagic dilapidation and autophagosome formation [56,57]. Beclin-1 is a key regulator of autophagy and exhibited increased expressions in the PD model [58]. Loss of the mitochondrial membrane potentially facilitates the buildup of p62 in clustered mitochondria in a PARKIN-dependent manner. In line with these reports, our results showed an upsurge of beclin-1, LC3, and p62 in ROT-challenged rats, indicating impairment in autophagic flux; however, myrcene abolished the adverse effects of ROT by normalizing the expression of autophagic markers. The depletion of nutrients in cell homeostasis is mediated by the mTOR signaling pathway, which is more likely macroautophagy. During neuronal growth, mTOR is required for lipid synthesis, which is an important factor in cell membrane formation and other functions. 

In addition, mTOR participates in maintaining the shape, migration, and differentiation of neuronal cells, and is thus important for the development of memory, and sustaining synaptic plasticity. Inhibition of mTOR has been shown to be associated with impairment of p70S6 kinase and 4E-BP-1, thereby promoting impaired homeostasis in neuronal growth. Hence, the expressions of mTOR, p-mTOR, and p70S6 kinase were analyzed in the present study [26,59,60,61,62]. ROT diminished the expression of mTOR, p-mTOR, and p70S6 kinase, thereby impairing cell growth and homeostasis. However, myrcene restored the expression of mTOR, p-mTOR, and p70S6 kinase, facilitated the maintenance of homeostasis, and restored cell growth and survival. 

Collectively, the present study findings demonstrate that ROT injections induce oxidative stress, inflammation, and apoptosis, along with an increase in α-synuclein and altered autophagy. However, myrcene treatment mitigates the aberrant increased expression of α-synuclein, diminishes the increased levels of pro-inflammatory cytokines and inflammatory mediators, decreases oxidative stress, and favorably alters autophagic flux, and mTOR signaling eventually results in the preservation of dopaminergic neurons. Given its dietary availability, accessibility, safety, and observed efficacy, myrcene could be an important agent for its therapeutic potential in PD, and can be suggested for its nutritional and phytomedicinal uses in neurodegenerative diseases.

## 4. Materials and Methods 

### 4.1. Experimental Animals and Ethics Approval

Experiments and experimental procedures were approved and performed in accordance with the guidelines of the Animal Ethics Committee of the United Arab Emirates University (UAEU). Male Wistar albino rats weighing 280–300 g were included in the present study. The rats were maintained under standard animal house conditions at 22 ± 1 °C, 50–55% humidity, and a 12/12 h photoperiod. 

### 4.2. Chemicals and Reagents

High-purity ROT and myrcene were obtained from Sigma-Aldrich (Missouri, MO, USA). Radioimmunoprecipitation assay (RIPA) buffer, antibodies for inducible nitric oxide synthase (iNOS), cyclooxygenase-2 (COX-2), and glial fibrillary acidic protein (GFAP) were obtained from Sigma-Aldrich (Missouri, MO, USA). Protease and phosphatase inhibitor cocktails were purchased from Thermo Fisher Scientific (Waltham, MA, USA). The anti-tyrosine hydrolase (polyclonal rabbit) antibody was purchased from Merck (Darmstadt, Germany). LC3, p62, mTOR, phospho-mTOR, and p70S6 antibodies were obtained from Cell Signaling Technology (Danvers, MA, USA). The apoptotic polyclonal markers, Bax and Bcl-2, were purchased from Abcam (Cambridge, MA, USA). The monoclonal mouse anti-α-synuclein antibody was purchased from BD Biosciences (San Jose, CA, USA). Anti-ionized calcium-binding adapter molecule 1 (Iba-1) antibody was procured from Wako Chemicals (Richmond, VA, USA). Alexa Fluor 488, a fluorescent secondary antibody, was obtained from Thermo Fisher Scientific (Waltham, MA, USA). Biotinylated goat anti-rabbit secondary antibody was procured from Jackson Immune Research Laboratories (Baltimore Pike, West Grove, PA, USA). Biochemical assays were performed using kits available commercially. All other analytical grade chemicals used in these experiments were procured from local distributors. 

### 4.3. Study Design and Experimental Protocol

PD was induced in experimental rats by administering ROT injections (2.5 mg/kg, i.p.), as standardized in our laboratory and as described previously [13,14,15]. A dose-dependent study was performed with myrcene (25, 50 and 100 mg/kg body weight, orally) to reveal its dose dependent effect against ROT-induced neurotoxicity in rats (Figure 1). We observed that myrcene treatment (50 and 100 mg/kg body weight, orally) considerably decreased the altered oxidative stress markers (MDA and GSH) in ROT-induced rats. Since myrcene (50 mg/kg body weight, orally) showed the highest effect, we fixed this dose for our further studies. ROT was dissolved in dimethyl sulfoxide (DMSO) to prepare the stock solution, which was further diluted in mygliol to reach a final concentration of 2.5 mg/mL. Rats were divided into four groups (*n* = 15). Group I (control): Rats received olive oil, DMSO in mygliol alone in an amount equal to myrcene, and ROT-alone treatment; Group II: Rats were treated with ROT (2.5 mg/kg, i.p.) along with olive oil (similar amount used for dissolving myrcene); Group III: Rats were orally treated with myrcene (50 mg/kg) dissolved in olive oil and DMSO in mygliol; Group IV: Rats were pretreated with myrcene (50 mg/kg), followed by ROT (2.5 mg/kg, i.p.). In all groups, treatments were carried out 5 days a week for a period of 28 days.

### 4.4. Tissue Handling and Sample Preparation

Following anesthesia with pentobarbital (40 mg/kg body weight), the animals underwent intracardiac perfusion with phosphate buffered saline (PBS) (0.01 M). For immunofluorescence analysis, the rats were perfused with PBS (0.01 M), followed by 4% paraformaldehyde. The brain was aseptically removed, and the midbrain and striatum were extracted for further experiments. Whole brains were incubated in 4% paraformaldehyde for 48 h, followed by sucrose treatment for 3 consecutive days. The midbrain was homogenized in KCl buffer containing 10 mM Tris-HCl, 140 mM NaCl, 300 mM KCl, 1 mM ethylenediaminetetraacetic acid, and 0.5% Triton-X100 at pH 7.5, following the addition of protease and phosphatase inhibitors. Tissues were centrifuged at 13,000 rpm for 30 min at 4 °C. The supernatants were analyzed using ELISA and biochemical assays. Striatum tissues were processed using RIPA buffer for Western blot analysis.

### 4.5. Estimation of Malondialdehyde (MDA) Assay

MDA assay was performed to evaluate lipid peroxidation using the North-West Life Science MDA detection kit (Vancouver, WA, USA), following the manufacturer’s instructions. Sample or calibrators (100 μL), thiobarbituric acid (100 μL), BHT (5 μL), and acid reagent (100 μL) were dispensed in an Eppendorf tube. The samples were incubated at 60 °C for 1 h, followed by centrifugation for 2–3 min at 2000 rpm. The reaction mixture was transferred to a 96-well ELISA plate, and the absorbance was read at 532 nm. The absorbance data were processed, and the results were calculated and presented as micromolar MDA.

### 4.6. Estimation of Glutathione (GSH)

GSH content was analyzed using a commercially available GSH assay kit (Sigma-Aldrich Chemie GmbH, Stein Heim, Germany) in all experimental groups. Briefly, the samples were deproteinized using 5% 5-sulfosalicylic acid, and 10 μL of sample or standard was dispensed in a 96-well plate, followed by 150 μL of a working mixture consisting of assay buffer, 5,5′-dithiobis (2-nitrobenzoic acid), and GSH reductase. NADPH (50 µL) was added to the reaction mixture for optimal color development, and kinetic readings were recorded for 6 min at 412 nm. GSH content was calculated and expressed as micromolar GSH. 

### 4.7. Determination of Antioxidant Enzymes; Superoxide Dismutase (SOD) and Catalase

SOD and catalase activities were estimated in brain samples using commercial kits procured from Cayman Chemicals Company (Ann Arbor, MI, USA), following the manufacturer’s protocols. For the catalase assay, 20 μL of the sample or standard, 30 μL of methanol, and 100 μL of assay buffer were dispensed in a 96-well plate. The reaction was initiated by adding 20 μL of hydrogen peroxide (H_2_O_2_) and incubating for 20 min. Next, 30 μL of KOH was added to terminate the reaction, and 30 μL of catalase purpald was added for optimal color development to complete the reaction. Catalase potassium periodate (10 μL) was then added to stop the reaction. Absorbance was recorded at 540 nm and catalase activity was calculated in nM/mL/min. For SOD estimation, 10 μL of sample or standard was added to a 96-well microplate, followed by 20 μL of xanthine oxidase to start the reaction. The microplate was incubated for 30 min at room temperature. Thereafter, absorbance was read at 450 nm and SOD activity was recorded in U/mL.

### 4.8. Estimation of Pro-Inflammatory Cytokines and MMP-9

The concentrations of pro-inflammatory cytokines (TNF-α, IL-1β, and IL-6) and MMP-9 were measured in midbrain tissues using commercially available kits (R&D Systems, Minneapolis, MN, USA). The plates were coated with primary antibody in PBS (0.01 M, pH 7.5) and incubated overnight. Next, the plates were washed thrice, blocked with 300 μL of 1% bovine serum albumin (BSA), and incubated for 1 h. The plates were washed thrice, and 100 μL of the pre-prepared standard and samples were added to the plates, and incubated for 2 h. After incubation, the plates were washed thrice, and 100 μL of secondary antibody was added, followed by 2 h of incubation. Subsequently, the plates were washed thrice, and 100 μL of HRP was added, again washed thrice, and 100 μL of TMB was added to develop the color. The colorimetric reaction was stopped by adding 50 μL of 2N sulfuric acid. Absorbance was recorded at 450 nm, and the concentrations of pro-inflammatory cytokines and MMP-9 were calculated as pg/mL. 

### 4.9. Estimation of Microglia and Astrocyte Activation by Immunofluorescence Staining

The striatal sections of all experimental groups were used to stain astrocytes (GFAP) and microglia (Iba-1). Sections with 30 µm thickness were washed twice in a 24-well tissue plate using the floating section method. Briefly, striatum tissue sections were blocked with a blocking reagent consisting of 10% normal goat serum in PBS, 0.3% Triton-X100, and 1% BSA for 1 h. The sections were incubated overnight with primary antibodies GFAP (1:1000) and Iba-1 (1:1000) at 4 °C representing the expression of astrocytes and microglia, respectively. The sections were washed twice with PBS and incubated with the corresponding fluorescent secondary antibody (Alexa 488) (anti-rabbit or anti-mouse) specific to the primary antibody. The sections were incubated for 1 h and washed twice with 1x PBS, and mounted on slides. The sections on slides were mounted using fluorescent mounting media and Vecta stained with DAPI. Photomicrographs were visualized under a fluorescence microscope (Nikon Eclipse Ni). From each brain, a minimum of three coronal sections at a similar level of the striatum were used to analyze the number of activated microglial cells and astrocytes. After random selection of three different fields of equal area, the images were analyzed using the ImageJ software (NIH, Bethesda, MD, USA). An outline was drawn around the region of interest, and the area, circularity, and mean fluorescence were determined considering numerous adjacent background readings. Total corrected cellular fluorescence (TCCF) was calculated using the formula: TCCF = integrated density - area of selected cell × mean fluorescence of background readings). All readings were determined by an observer blinded to the experimental protocols to avoid any bias in the calculation. Data are presented as percentages of the control.

### 4.10. Western Blotting for Proteins

The striatal tissues were used for Western blotting to determine the expression of specific proteins. Western blotting was performed as described previously (13, 14, 15). Striatal tissues were homogenized in RIPA buffer supplemented with phosphatase and protease inhibitors. Proteins (35 µg) from each sample were loaded and separated using SDS-PAGE. Proteins from the gels were electro-transferred onto a polyvinylidene fluoride membrane using the trans-turbo Bio-Rad method. The membrane was blocked with 5% nonfat dry milk in PBS with 0.1% Tween 20 (PBST) for 1 h. After blocking, the membranes were washed with PBST and incubated overnight at 4 °C with the following primary antibodies: α-synuclein (1:750), COX-2 (1:2000), iNOS (1:1000), Bax (1:2000), Bcl-2 (1:500), mTOR (1:1500), phospho-mTOR (1:900), p62 (1:900), LC3 (1:800), and p70S6K (1:900). The membranes were washed with PBST, and secondary antibodies specific to the primary antibodies were added to the membrane and incubated at room temperature for 1 h. The membranes were developed for protein bands using a Chemiluminescence Pico Kit Thermo Fisher Scientific, (Waltham, MA, USA). The membranes were stripped and re-probed for β-actin to determine the total protein content. The observed bands were quantified using the ImageJ software.

### 4.11. Assessment of Tyrosine Hydroxylase (TH-Staining) for Dopaminergic Neurons in SN and Striatum

Cryoprotected rat brains were serially sectioned (25 µm) using a cryostat (Leica, Wetzlar, Germany). Serial sections of the SN were washed twice using PBS. The sections were inactivated for peroxidase using 1% H_2_O_2._ The samples were again washed with PBS twice and blocked using blocking buffer (10% normal goat serum in PBS containing 0.3% Triton-X 100) for 1 h. After blocking, sections were washed with 1x PBS, and primary antibody was added (goat anti-rabbit tyrosine polyclonal antibody, 1:1000) overnight at 4 °C. The sections were washed twice with 1% PBS, and biotinylated secondary antibody (1:1000) was dispensed and incubated for 1 h. The sections were again washed twice and developed with an avidin–biotin peroxidase complex system (ABC kit, Vectastain, Newark, CA, USA) followed by 3,3′ diaminobenzidine (DAB) to visualize and examine TH immunoreactivity. Sections were then mounted on slides with DPX media and cover slips, examined under a light microscope, and representative images were analyzed. 

### 4.12. Assessment of TH-ir Dopaminergic Neurons and TH-ir Dopamine Nerve Fibers Loss

The loss of nerve fibers in the SN area was analyzed. The stained sections were examined for loss of TH-immunoreactive (TH-ir) neurons. Briefly, sections at three to five different levels of the medial terminal nucleus region were counted. The counts are represented as a percentage of control to naive cells [13,14,15]. Striatal fiber loss was analyzed by calculating the optical density of TH-ir dopaminergic fibers in the striatum using the ImageJ software. The optical densities of TH-ir from three different sections were obtained from each group, and the measured density was represented as a percentage of the naïve control. 

### 4.13. Protein Estimation

The protein content in each sample was quantified using the Pierce BCA protein assay kit Thermo Fisher Scientific, (Waltham, MA, USA) following the manufacturer’s protocol.

### 4.14. Statistical Analysis

The data were analyzed using one-way analysis of variance followed by Tukey’s test. The criterion of statistical significance between different experimental groups was established by employing the SPSS12 software. Values are expressed as mean ± standard error of the mean (SEM). *p* < 0.05 was considered statistically significant.

## Figures and Tables

**Figure 1 molecules-28-00685-f001:**
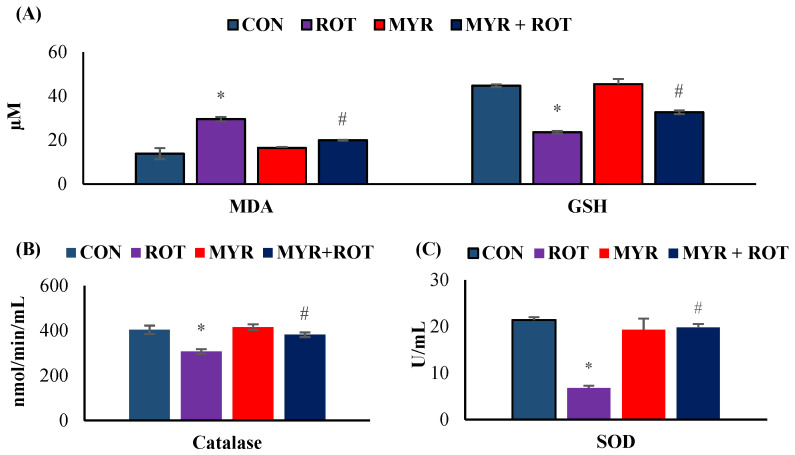
The effect of myrcene on oxidative stress markers MDA and GSH (**A**) and antioxidant defense markers CAT and SOD (**B**,**C**). The values are expressed as mean ± SEM (*n* = 6–8). * *p* < 0.05 (ROT vs. CON), ^#^
*p* < 0.05 (MYR vs. ROT).

**Figure 2 molecules-28-00685-f002:**
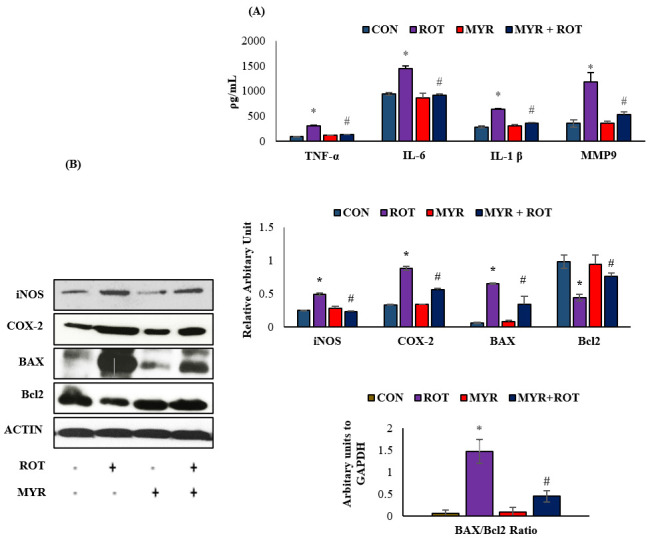
(**A**)**.** The effect of myrcene on the levels of pro-inflammatory cytokines; TNF-α, IL-6, IL-1β, and MMP-9. The values are expressed as mean ± SEM (*n* = 6–8). * *p* < 0.05 (ROT vs. CON), ^#^
*p* < 0.05 (MYR vs. ROT). (**B**). The effect of myrcene on the expression of inflammatory enzymes (iNOS and COX-2) and apoptotic markers (BAX, Bcl-2 and BAX/Bcl-2 ratio). The values are expressed as percent mean ± SEM (*n* = 3). * *p* < 0.05 (ROT vs. CON), ^#^
*p* < 0.05 (MYR vs. ROT).

**Figure 3 molecules-28-00685-f003:**
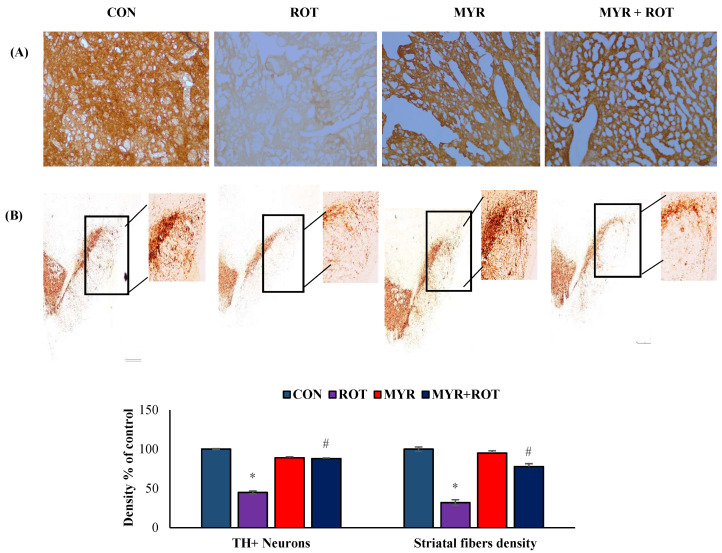
The effect of myrcene on the expression of TH-ir dopamine nerve fibers in the striatum (**A**) and number of dopaminergic (DA) neurons in the substantia nigra compacta (SN) (**B**). Scale bar: 100 µm. The values are expressed as percent mean ± SEM (*n* = 3). * *p* < 0.05 (ROT vs. CON), ^#^
*p* < 0.05 (MYR vs. ROT).

**Figure 4 molecules-28-00685-f004:**
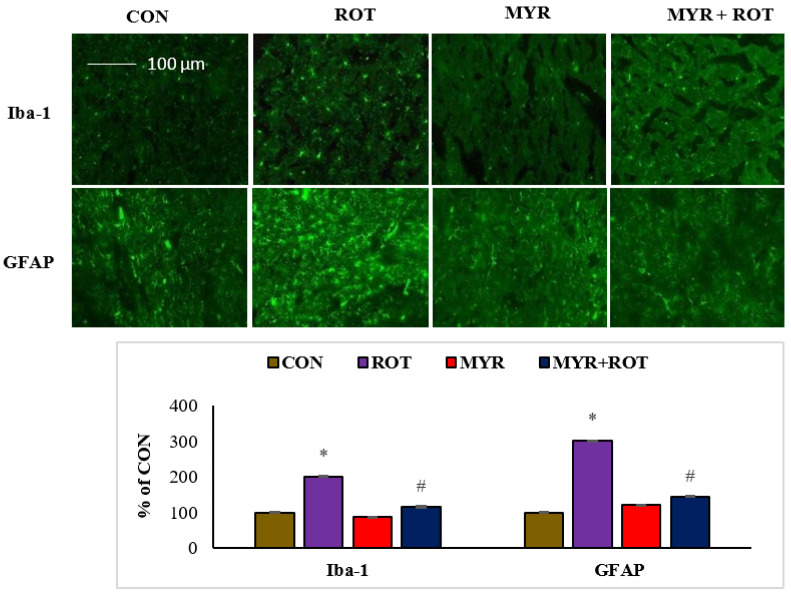
The effect of myrcene on activation of glial fibrillary acidic protein (GFAP) positive astrocyte (green) and ionized calcium binding adaptor molecule-1 (Iba-1) positive microglia (green) in the striatum. Scale bar: 100 µM. The values are expressed as percent mean ± SEM (*n* = 3). * *p* < 0.05 (ROT vs. CON), ^#^
*p* < 0.05 (MYR vs. ROT).

**Figure 5 molecules-28-00685-f005:**
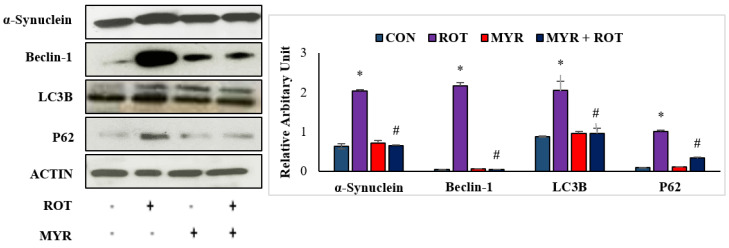
The effect of myrcene on the expression of α-synuclein, a protein playing a role in PD, and autophagy markers; Beclin-1, LC3B, and p62. The values are expressed as percent mean ± SEM (*n* = 3). * *p* < 0.05 (ROT vs. CON), ^#^
*p* < 0.05 (MYR vs. ROT).

**Figure 6 molecules-28-00685-f006:**
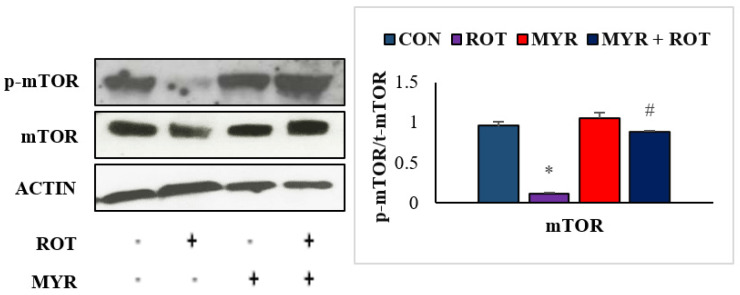
The effect of myrcene on expression of mTOR and p-mTOR. The values are expressed as percent mean ± SEM (*n* = 3). * *p* < 0.05 (ROT vs. CON), ^#^
*p* < 0.05 (MYR vs. ROT).

## Data Availability

The data will be available on reasonable request.

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
