# Peer review of "Myrcene Salvages Rotenone-Induced Loss of Dopaminergic Neurons by Inhibiting Oxidative Stress, Inflammation, Apoptosis, and Autophagy"

_molecules, 2023, doi:10.3390/molecules28020685_

Round 1
Reviewer 1 Report
The current report is an interesting novel study about how the phytochemical myrcene is highly protective against numerous pathopshysiologic traits of Parkinson's disease experimental model by rotenone. Overall, the experiments have been designed and performed correctly, the text is well-written and the conclusion is clear.
I have small issues to be solved before acceptance.
Material and Methods
Why only male animals were used?
The dose used for myrcene came from a previous study? A reference should be added or an explanation of why was this dose chosen.
I appreciate that myrcene was given orally; however, there is not a discussion about how much of myrcene would reach the brain. Ideally, plasma concentration levels should be measured. If not, I would be ok with information from the literature discussing this subject.
Statistics are not appropriately described or analyzed. For instance, all the experiments using the four experimental groups where there is drug association, a two-way ANOVA analysis should be performed rather than a one-way ANOVA. Also, no ANOVA result is presented throughout the text (F values), rather than only post-hoc comparisons.
Results
Figure 3 appears to be in low quality, it is hard to determine if TH+ loss is for sure a loss or just reducing of TH staining, specially in substantia nigra (which could provide different information about TH-reduced staining vs cell death). I would suggest a higher magnification of substantia nigra photomicrographs to show neuronal bodies in control and treated groups.
In figure 4, it is hard to confirm A and B are respectively Iba-1 and GFAP. This should be shown in figure legend.
In result 6, there are numerous conflicting data about ROT being able to induce alpha-synuclein aggregation. Is it possible to affirm that the alpha-synuclein that is increased by ROT is aggregated? It would be interesting to confirm whether this increase in alpha-synuclein immunoreaction is indeed aggregated in your experimental procedure, and that myrcene is avoiding aggregation.
Discussion
“Collectively, the present study found in line with previous findings that increase in α-synuclein aggregation is a default mechanism after ROT-insult.” – as stated above, it is not appropriate to affirm that your results demonstrate it, as you are not showing specific reaction to alpha-synuclein fibrils or oligomers.
Author Response
Reviewer-1
Material and Methods
- Why only male animals were used?
Reply: Male rats were used as the incidence of PD male-female ratio is 1.5 to 1 for rotenone. However, PD is reported in both male and female rats De Mirinda et al., 2019]. We used male rats for our experiments due to its easy availability.
Reference:
De Miranda, B.R.; Fazzari, M.; Rocha, E.M.; Castro, S.; Greenamyre, J.T. Sex Differences in Rotenone Sensitivity Reflect the Male-to-Female Ratio in Human Parkinson's Disease Incidence. Toxicol Sci 2019, 170, 133-143, doi:10.1093/toxsci/kfz082.
- The dose used for myrcene came from a previous study. A reference should be added or an explanation of why was this dose chosen.
Reply: The dose of myrcene was selected based on dose dependent study. We included the details in the revised manuscript.
- Statistics are not appropriately described or analyzed. For instance, all the experiments using the four experimental groups where there is drug association, a two-way ANOVA analysis should be performed rather than a one-way ANOVA. Also, no ANOVA result is presented throughout the text (F values), rather than only post-hoc comparisons.
Reply: We use one-way ANOVA and p values were (p < 0.05) considered significant. We will consider your valuable consideration on running two-way analysis to show the statistical significance in future.
- Results
Figure 3 appears to be in low quality, it is hard to determine if TH+ loss is for sure a loss or just reducing of TH staining, especially in substantia nigra (which could provide different information about TH-reduced staining vs cell death). I would suggest a higher magnification of substantia nigra photomicrographs to show neuronal bodies in control and treated groups.
Reply: We replaced the figures for TH+ staining in the revised manuscript.
- In figure 4, it is hard to confirm A and B are respectively Iba-1 and GFAP. This should be shown in figure legend.
Reply: We did the changes as per your suggestion.
- In result 6, there are numerous conflicting data about ROT being able to induce alpha-synuclein aggregation. Is it possible to affirm that the alpha-synuclein that is increased by ROT is aggregated? It would be interesting to confirm whether this increase in alpha-synuclein immunoreaction is indeed aggregated in your experimental procedure, and that myrcene is avoiding aggregation.
Reply: Many thanks for your comments. Rotenone is very well known to induce α-synuclein aggregation in animal models [Yuan et al., 2015]. Unfortunately, due to the unavailability of samples, we can't able to perform Immunohistochemistry on α-synuclein aggregation. As per your valuable suggestion, we will do immunostaining to reveal α-synuclein aggregation in our future studies.
Reference
Yuan, Y.H.; Yan, W.F.; Sun, J.D.; Huang, J.Y.; Mu, Z.; Chen, N.H. The molecular mechanism of rotenone-induced α-synuclein aggregation: emphasizing the role of the calcium/GSK3β pathway. Toxicol Lett 2015, 233, 163-171, doi:10.1016/j.toxlet.2014.11.029.
- Discussion
“Collectively, the present study found in line with previous findings that increase in α-synuclein aggregation is a default mechanism after ROT-insult.” – as stated above, it is not appropriate to affirm that your results demonstrate it, as you are not showing specific reaction to alpha-synuclein fibrils or oligomers.
Reply: We removed the statement in the discussion section. We will consider your valuable suggestion in including the immunostaining analysis to reveal the synuclein fibrils or oligomers in the future.
Reviewer 2 Report
The authors characterize antioxidant and anti-inflammatory properties of myrcene and its effect on apoptosis and autophagy markers using a neurotoxic model of Parkinson disease induced by rotenone. Interestingly, myrcene was able to protect dopaminergic neurons against rotenone toxicity by reducing TH loss and attenuating the activation of glia. Other strong points of this work are the evaluation of oxidative and inflammatory markers in this animal model of PD pointing towards an antioxidant and antiinflamatory effect of myrcene.
In the following lines, some weaknesses and potential improvements of this work will be pointed out.
Major
· Western blot samples were obtained from the striatum, therefore protein expression is not directly related to dopaminergic neurons.
· Western blot of iNOS seems to be not correlated with the total actin protein as band is up in the fourth lane.
· Authors did not evaluate aggregation of α-synuclein (only expression by Western blot).
· Regarding autophagy analysis: LC3B was analyzed as ratio LC3II/LC3I but the Western blot does not show two bands. Beclin-1 expression is not discussed in the Results section.
· mTOR signaling is involved in other cellular events, such as autophagy. To conclude that this pathway is associated with myrcene effect on ROT-induced apoptosis, further experiments should be done using, for example, an inhibitor of mTOR and subsequent determination of apoptotic markers in the presence of myrcene.
Minor
· Figures 2 and 5 provide complementary data so they should be included in a single figure.
· For apoptotic markers, it is better to express results as Bax/Bcl2 ratio.
· What do you mean with behavioral toxicity?
Author Response
Reviewer 2
- Western blot samples were obtained from the striatum; therefore, protein expression is not directly related to dopaminergic neurons.
Reply: The striatum sample were used due to its precise isolation, however the substantia nigra isolation and the size of the sample are having some technical issues. Therefore, for biochemical we use midbrains having substantia nigra and the results are shown. Second the loss of striatal dopaminergic terminal first appears (70%) followed by the substantia dopaminergic neurons (30-40%).
- Western blot of iNOS seems to be not correlated with the total actin protein as band is up in the fourth lane.
Reply: We replaced the iNOS band in the revised manuscript.
- Authors did not evaluate aggregation of α-synuclein (only expression by Western blot).
Reply: Many thanks for your comments. Rotenone is very well known to induce α-synuclein aggregation in animal models [Yuan et al., 2015]. Unfortunately, due to the unavailability of samples, we can't able to perform Immunohistochemistry on α-synuclein aggregation. We believe that the immunoblotting analysis on α-synuclein is sufficient to prove the mechanism. As per your valuable suggestion, we will do immunostaining to reveal α-synuclein aggregation in our future studies.
- Regarding autophagy analysis: LC3B was analyzed as ratio LC3II/LC3I but the Western blot does not show two bands. Beclin-1 expression is not discussed in the Results section.
Reply: We replaced the LC3B results in the revised manuscript.
- mTOR signaling is involved in other cellular events, such as autophagy. To conclude that this pathway is associated with myrcene effect on ROT-induced apoptosis, further experiments should be done using, for example, an inhibitor of mTOR and subsequent determination of apoptotic markers in the presence of myrcene.
Reply: Many thanks for your suggestion. Unfortunately, due to the unavailability of samples we are unable to perform the suggested experiments using mTOR inhibitor since we finished this project before two years. We will consider your valuable suggestion for our future experiments.
- Figures 2 and 5 provide complementary data so they should be included in a single figure.
Reply: We modified figure 2 and 5 into a single figure as suggested by the reviewer.
- For apoptotic markers, it is better to express results as Bax/Bcl2 ratio.
Reply: We included Bax/Bcl2 ratio in the revised manuscript.
- What do you mean with behavioral toxicity?
Reply: We corrected the statement in the revised manuscript.
Round 2
Reviewer 2 Report
- Western blot samples were obtained from the striatum; therefore, protein expression is not directly related to dopaminergic neurons.
Reply: The striatum sample were used due to its precise isolation, however the substantia nigra isolation and the size of the sample are having some technical issues. Therefore, for biochemical we use midbrains having substantia nigra and the results are shown. Second the loss of striatal dopaminergic terminal first appears (70%) followed by the substantia dopaminergic neurons (30-40%).
Reviewer 2 reply: I understand the point that substantia nigra (SN) is not easy to isolate, of course. Then, it is wrong from a scientific view to detail that the results were obtained from SN. So, to be more accurate, authors have to change SN for striatum in the text.
- Western blot of iNOS seems to be not correlated with the total actin protein as band is up in the fourth lane.
Reply: We replaced the iNOS band in the revised manuscript.
Reviewer 2 reply: Ok.
- Authors did not evaluate aggregation of α-synuclein (only expression by Western blot).
Reply: Many thanks for your comments. Rotenone is very well known to induce α-synuclein aggregation in animal models [Yuan et al., 2015]. Unfortunately, due to the unavailability of samples, we can't able to perform Immunohistochemistry on α-synuclein aggregation. We believe that the immunoblotting analysis on α-synuclein is sufficient to prove the mechanism. As per your valuable suggestion, we will do immunostaining to reveal α-synuclein aggregation in our future studies.
Reviewer 2 reply: One band in western blot using alpha synuclein antibody is not enough prove to demonstrate aggregation of the protein. Indeed, when alpha synuclein suffers aggregation (it could happen in the rotenone model and others scenarios of neurotoxicity), multiple bands are detected by western blot corresponding to the oligomers of alpha synuclein. Therefore, I do not agree with authors that immunoblotting analysis on α-synuclein is sufficient to prove aggregation. They have to change aggregation of alpha synuclein by the more precise term “expression of alpha synuclein”.
- Regarding autophagy analysis: LC3B was analyzed as ratio LC3II/LC3I but the Western blot does not show two bands. Beclin-1 expression is not discussed in the Results section.
Reply: We replaced the LC3B results in the revised manuscript.
Reviewer 2 reply: The replacement of LC3B indicates now the results described in the manuscript. However, beclin-1 is still missed in the text (but authors showed the western blot and quantification). Please include it.
alpha synuclein
- mTOR signaling is involved in other cellular events, such as autophagy. To conclude that this pathway is associated with myrcene effect on ROT-induced apoptosis, further experiments should be done using, for example, an inhibitor of mTOR and subsequent determination of apoptotic markers in the presence of myrcene.
Reply: Many thanks for your suggestion. Unfortunately, due to the unavailability of samples we are unable to perform the suggested experiments using mTOR inhibitor since we finished this project before two years. We will consider your valuable suggestion for our future experiments.
Reviewer 2 reply: I understand the limitation of performing new experiments. Then, you can not connect apoptosis with mTOR signaling. So, please, limit the interpretation of the results to mTOR signaling but not to its correlation with apoptosis because this pathway participates in other cellular events,
Figures 2 and 5 provide complementary data so they should be included in a single figure.
Reply: We modified figure 2 and 5 into a single figure as suggested by the reviewer.
Reviewer 2 reply: Ok.
- For apoptotic markers, it is better to express results as Bax/Bcl2 ratio.
Reply: We included Bax/Bcl2 ratio in the revised manuscript.
Reviewer 2 reply: Ok.
- What do you mean with behavioral toxicity?
Reply: We corrected the statement in the revised manuscript.
Reviewer 2 reply: Ok.
Author Response
Many thanks for your positive consideration and suggestions to improve the manuscript. We sincerely appreciate your efforts, and valuable time in reviewing our manuscript.
Reviewer 2: I understand the point that substantia nigra (SN) is not easy to isolate, of course. Then, it is wrong from a scientific view to detail that the results were obtained from SN. So, to be more accurate, authors have to change SN for striatum in the text.
Reply: As advised, we did the changes in the text in the revised manuscript.
--------------------------------------------------------------------------
Reviewer 2: One band in western blot using alpha synuclein antibody is not enough prove to demonstrate aggregation of the protein. Indeed, when alpha synuclein suffers aggregation (it could happen in the rotenone model and others scenarios of neurotoxicity), multiple bands are detected by western blot corresponding to the oligomers of alpha synuclein. Therefore, I do not agree with authors that immunoblotting analysis on α-synuclein is sufficient to prove aggregation. They have to change aggregation of alpha synuclein by the more precise term “expression of alpha synuclein”.
Reply: We appreciate the reviewer's remarks. We fully agree with the reviewer and as advised, modified the text in the revised manuscript.
-------------------------------------------------------------------------
Reviewer 2: The replacement of LC3B indicates now the results described in the manuscript. However, beclin-1 is still missed in the text (but authors showed the western blot and quantification). Please include it.
Reply: Many thanks for your comments. As advised, beclin-1 has been included in the revised manuscript.
-----------------------------------------------------------------------------
Reviewer 2: I understand the limitation of performing new experiments. Then, you can not connect apoptosis with mTOR signaling. So, please, limit the interpretation of the results to mTOR signaling but not to its correlation with apoptosis because this pathway participates in other cellular events,
Reply: We do agree with the reviewer's remark and thanks for this comment. As advised, we limited the interpretation on linking apoptosis and mTOR signaling. -------------------------------------------------------------------------
We thank you once again for your positive consideration and for providing an opportunity to improve it and make it interesting and appealing.